# Proteomics as a New-Generation Tool for Studying Moulds Related to Food Safety and Quality

**DOI:** 10.3390/ijms24054709

**Published:** 2023-03-01

**Authors:** Micaela Álvarez, María J. Andrade, Félix Núñez, Mar Rodríguez, Josué Delgado

**Affiliations:** Higiene y Seguridad Alimentaria, Instituto Universitario de Investigación de Carne y Productos Cárnicos, Facultad de Veterinaria, Universidad de Extremadura, Avda. de las Ciencias s/n, 10003 Cáceres, Spain

**Keywords:** foodborne moulds, proteomics, food safety, food quality, detection, mechanisms of action, antifungal agents

## Abstract

Mould development in foodstuffs is linked to both spoilage and the production of mycotoxins, provoking food quality and food safety concerns, respectively. The high-throughput technology proteomics applied to foodborne moulds is of great interest to address such issues. This review presents proteomics approaches useful for boosting strategies to minimise the mould spoilage and the hazard related to mycotoxins in food. Metaproteomics seems to be the most effective method for mould identification despite the current problems related to the bioinformatics tool. More interestingly, different high resolution mass spectrometry tools are suitable for evaluating the proteome of foodborne moulds able to unveil the mould’s response under certain environmental conditions and the presence of biocontrol agents or antifungals, being sometimes combined with a method with limited ability to separate proteins, the two-dimensional gel electrophoresis. However, the matrix complexity, the high ranges of protein concentrations needed and the performing of multiple steps are some of the proteomics limitations for the application to foodborne moulds. To overcome some of these limitations, model systems have been developed and proteomics applied to other scientific fields, such as library-free data independent acquisition analyses, the implementation of ion mobility, and the evaluation of post-translational modifications, are expected to be gradually implemented in this field for avoiding undesirable moulds in foodstuffs.

## 1. Introduction

Moulds are a key microbial group in the food industry, since they are capable of growing in a wide range of environmental conditions. Firstly, the application of moulds and derived products to produce and preserve food and food ingredients is very broad [1]. Mould enzymes are ubiquitous, used in starch processing, in the bakery, and brewery industries, and to produce beverages, including wines and in food fermentation [1]. Apart from producing beneficial effects, moulds are the most commonly found spoilage microorganisms at every stage of the food chain and could be the primary causes of significant financial losses in some foodstuffs [2]. Additionally, this microbial group poses issues to human health because of their potential production of undesirable compounds, especially mycotoxins. Both harmful activities linked to mould development are a concern in the food industry since they can seriously damage the brand image [3].

During the period 2018-2022 a total of 131 notifications related to mould contamination of different animal and vegetal food products and food supplements were accepted in the European Union [4]. Within these notifications, 1 was classified as “alert”, 42 as “border rejection”, and 88 as “information”. The highest occurrence of moulds in such products was declared in cereals and bakery products. Regarding mycotoxin notifications during this period, more than 1000 have been stated, with aflatoxins being the most frequently found, followed by ochratoxin A [4]. Most of the notifications concerned the categories “nuts, nut products and seeds”, “cereals and bakery products”, “herbs and spices”, and “fruits and vegetables”. 

To guarantee the quality and safety of foodstuffs in relation to undesirable moulds, different techniques for their detection and the characterisation of their mechanisms of action have been reported. The quick development of modern technology has encouraged the application of omics for such a purpose. In the last decades, an increasing number of proteomics approaches have been proposed to be used specially to discover the key mechanisms of action of moulds with interest in foodstuffs. This omics technique has some advantages when compared to other techniques working with the same aim, such as transcriptomics. Thus, the primary RNA transcript of eukaryotic genes can be processed in more than one way, resulting in more than one protein from a single gene [5]. The rise in the use of this tool is the result of different advancements, for example, the increased number of available protein sequences, the technological developments with respect to the analysis of protein mixtures, and the improvement of bioinformatics tools to generate and process large biological data sets [1]. The proteomic methodology has been successfully utilised for investigating the microbe–host interactions, the pathogenic processes and toxin biosynthesis, and the responses of microorganisms to environmental factors [6,7,8,9,10,11,12,13,14]. Thus, proteomics could provide the knowledge for boosting strategies to avoid the issues caused by the mould spoilage and the hazard related to mycotoxins in food. 

This review presents approaches of the high-throughput technology proteomics applied to foodborne moulds for understanding how to address the food quality and safety challenges, showing their advantages and downsides. Apart from being an overview of the proteomics techniques currently available for achieving such a purpose, this work intends to provide knowledge about how they could be improved when applied to other scientific fields. 

## 2. Problems Associated with Moulds in Foodstuffs

As stated before, two downsides associated with the mould contamination of food are of interest: spoilage and mycotoxin production provoking food quality and food safety concerns, respectively. 

Regarding alteration, filamentous fungi are considered a severe pathogen of food due to their ability to penetrate and break down food components using extracellular enzymes [15]. They thus cause different types of spoilage, including unwanted visible mycelium on the product surface and undesirable sensory characteristics, such as flavour, colour, odour, and texture [16], with the consequent consumers’ rejection. *Penicillium*, *Aspergillus*, *Rhizopus*, *Mucor*, *Geotrichum*, *Fusarium*, *Alternaria*, *Cladosporium*, *Eurotium*, *Botrytis*, and *Byssochlamys* genera are involved in the spoiling of different foodstuffs [16,17]. Most of the problems related to mould spoilage have been described in fruits, vegetables, and grains and cereal products. For instance, bread and bakery products can rapidly spoil, mainly due to the growth of *Aspergillus*, *Penicillium*, *Rhizopus*, and *Mucor* species [18,19,20]. *Botrytis cinerea* is the main biological cause of pre- and post-harvest damage since it is responsible for grey mould formation in many plant species [17,21], including tomatoes [22] and table grapes [23]. Indeed, this undesirable mould is ranked second in the “world top 10 fungal pathogens in molecular plant pathology” in terms of economic and scientific relevance, preceded only by *Magnaporthe oryzae* [24]. Blue mould produced predominantly by *Penicillium expansum* and to a lesser extent other *Penicillium* spp. provokes the most detrimental infection of stored apples [25]. The white mould disease, caused by *Sclerotinia sclerotiorum*, is a major problem in rapeseed oil production [17]. Concerning food products of animal origin, black spot spoilage by moulds belonging to the *Cladosporium* genus (*Cladosporium oxysporum*, *C. cladosporioides*, and *C. herbarum*) has been reported in dry-cured ham and dry-cured fermented sausages [26,27]. *C. cladosporioides*, *C. herbarum*, *Penicillium hirsutum*, and *Aureobasidium pullulans* were isolated from chilled meat spoiled by black spot [28].

Considering the food safety issue associated with moulds, mycotoxins are a group of secondary metabolites with low molecular weight produced before and after the harvest of foodstuffs from vegetal origin and during ripening and the following processing of those from animal origin. In the latter, mycotoxin contamination could also be due to their presence in the animal feed [29,30]. These metabolites can provoke harmful effects, such as carcinogenic, immunosuppressive, teratogenic, and mutagenic ones [30] (Table 1). Hundreds of mycotoxins have been identified, but toxicity, frequency of outbreaks, and target organs differ among them [29]. Mycotoxin contamination is a great challenge to food safety since many of them cannot be eliminated using heat, physical, and chemical treatments [13,30]. 

The main mycotoxin-producing moulds belong to the genera *Fusarium*, *Aspergillus*, and *Penicillium*, which include several species producing toxins of the greatest concern worldwide, such as aflatoxins, ochratoxins, and fumonisins (Table 1). Other genera, such as *Claviceps*, *Alternaria*, etc., can also be involved [29] (Table 1). Other ones, such as *Fusarium* beauvericin, enniatins, and moniliformin, are so-called emerging mycotoxins and their serious risk on human and animal health have been stated despite the fact that a proper risk assessment has not been performed [30,36]. On the other hand, not every strain belonging to a mould species produce mycotoxins, and those that do usually produce them only in particular conditions [29].

Risks associated with mycotoxins depend on both hazard and exposure [37]. The hazard of mycotoxins to human beings is probably universal (while other factors are, occasionally, also important, for instance hepatitis B virus infection in relation to the hazard of aflatoxins). Exposure to mycotoxins is present worldwide; although, there are geographic and climatic differences in their production and occurrence as well as different dietary habits in various parts of the world [32,37]. However, the implication of global climate change in the toxigenic mould ecology and their pattern of mycotoxin production has been stated. As a result, the number of crops damaged by insects will increased because of global warming, and, therefore, render them more susceptible to mould infection [38], but it could also modify the diversity of diseases invading crops, certain mould might disappear from an environment and appear in new regions previously considered safe, along with the consequent economic and social implications [38]. Global warming will make crop growth impossible in some areas and, where growing crops will be possible, plants will be subjected to suboptimal climatic conditions, resulting in increased susceptibility to mould contamination [38]. Furthermore, warmer climates will favour thermotolerant species, leading to the prevalence of *Aspergillus* over *Penicillium* species [38]. Thus, climate change remains the primary factor for high levels of mycotoxins in African foods [39]. 

Many countries have regulated maximum limits and guidelines for relevant mycotoxins, such as aflatoxins, ochratoxin A, deoxynivalenol, zearalenone, fumonisins, T-2 toxin, HT-2 toxin, citrinin, Ergot sclerotia, ergot alkaloids, and patulin [32,40,41]. Current regulations are based on scientific opinions of authoritative bodies, such as the FAO/WHO Joint Expert Committee on Food Additives of the United Nations (JECFA) and the European Food Safety Authority (EFSA), which work with the known toxic effects [37].

Control of mould contamination is a major concern for the food industry and scientists that are looking for efficient solutions to prevent and/or limit not only their growth, but also their mycotoxin production. Chemical fungicides and good hygiene practices are the primary strategy for the treatment of undesirable foodborne moulds. Nonetheless, there is a growing demand from consumers for food free of synthetic fungicides and with a minor impact on the environment. Among the problems described for such products are the development of resistance to fungicides and the presence of residues in food, apart from causing allergies or side effects in some consumers [2,42]. As a result, major progress is being made in finding more sustainable and safer alternatives to such preservatives, including biopreservation, using microorganisms as well as legally permitted ingredients, and physical treatments. These alternatives generally do not have as wide a spectrum of activity as the synthetic fungicides [25] and, consequently, their combined application has been suggested [43,44,45]. 

Biological control using microorganisms have been reported for different food products. For instance, *Candida intermedia* provoked a significant reduction of ochratoxin A production when applied against *Aspergillus carbonarius* [46]. Both yeasts and bacteria have been investigated as biocontrol agents against grey mould decay in table grapes [23]. Biopreservation by lactic acid bacteria is considered the most promising alternative candidates to chemical fungicides in the dairy industry due to their Generally Regarded as Safe (GRAS) and Qualified Presumption of Safety (QPS) statuses in the United States and European Union, respectively [3,47]. Antimicrobial compounds of biological origin have also been investigated against undesirable foodborne moulds. Natural antimicrobials, including plant extracts, edible coating, and putrescine, amongst others, have been investigated against grey mould decay in table grapes [23]. Within plant extracts, essential oils from many plants have shown remarkable potential as biocontrol agents. Thus, numerous essential oils have been examined as antifungal agents for enhancing the shelf life of bread showing different degrees of impact; although the consumer does not always appreciate the flavour and aroma they provide [20]. For instance, tea tree oil (TTO) inhibited the spore germination and mycelial growth of *B. cinerea* [48]. Against such undesirable mould, the inhibitory biological effects of wuyiencin produced by *Streptomyces albulus* var. *wuyiensis* has also been reported [21].

## 3. Main Characteristics of Proteomics Applied to Moulds in Food 

Traditionally, the study of moulds’ transcripts has been employed to unveil the mould’s response under certain environmental conditions and the presence of biocontrol agents or antifungals [49,50,51,52,53]. However, the proteomic study is a more robust technique than transcriptomics due to proteins and not genes or transcripts being responsible for the cellular phenotypes. Indeed, gene expression alone does not provide information on post-translational modifications (PTM) or even protein expression itself, whilst proteomics offers the possibility to directly explore the expressed proteins [54]. These can be modified by the covalent attachment of substances, such as sugars, fats, phosphate groups, and others, affecting their performed function for the cell [5]. This could explain why some studies did not find a correlation between transcriptomics and proteomics in moulds. For instance, the changes in aflatoxigenic *Aspergillus flavus* protein profiles showed low congruency with alterations in the corresponding transcript levels, indicating that the post-translational processes play a critical role in regulating the protein level in this mould species [55]. Similarly, a proteomic investigation of *Aspergillus fumigatus* ∆gliT, related to gliotoxin production, did not reflect the large set of transcriptome changes [56]. Barker et al. [57] found decreased transcripts in abundance in two functional categories, glycolysis and amino acid metabolism, while the related proteins were enhanced in the spoilage mould *A. fumigatus.* A low correlation of transcriptome and proteome data was obtained in a toxigenic *A. flavus* grown in maize and peanut substrates [58]. Another study about *Tolypocladium guangdongense* used in medicine showed a low correlation between the transcriptomic and proteomic data, suggesting the importance of the post-transcriptional processes in its growth [59]. For all of them, together with the higher affordability of proteomics analyses in the last decades, it has emerged as the preferred omics tool in the study of the toxigenic mould physiology.

Despite the benefits of using this technique in food, it has some limitations as it depends on the matrix complexity, the high ranges of protein concentrations needed, and the performing of multiple steps [60]. To reduce some of these limitations, model systems have been used to simplify the experiments and avoid interferences with the food matrix and native microbial population. In parallel, and focusing on mould’s physiology, the model systems also facilitate the identification of mechanisms that could be hidden in a complex ecosystem [61]. Several studies have used commercial media to explore the mechanisms involved in the antifungal action of different compounds. For example, the ochratoxigenic *Aspergillus ochraceus* was grown in yeast extract sucrose broth to determine the influence of citral in its proteome [62]. The potato dextrose agar was used as substrate for the ochratoxin A-producing *A. carbonarius* growth in the presence of the volatile compound 2-phenylethanol [46] and *B. cinerea* treated with the antifungal wuyiencin [21], before protein extraction. Nevertheless, the use of food-based artificial media in proteomics allows for the reduction of contamination with proteins from the food itself or its native microbial population, bringing the experimental design closer to the real product than a commercial culture medium. In this sense, Xia et al. [7] employed an apple juice heated at 80 °C for 30 min to denature the present proteins in the juice as a food based-model system for *P. expansum* studies. Delgado et al. [63], also using apple as substrate, employed this fruit lyophilised together with agar and water to be sterilised, and subsequently, a layer of sterile cellophane was placed onto the solid medium to prevent cross contamination between proteins and *P. expansum.* Li et al. [58] used different broths made with crops powder substrates (maize, rice, and peanut) that were autoclaved before the inoculation of toxigenic *A. flavus*. A dry-cured fermented sausage-based medium was used to identify the modes of action of biocontrol agents against the ochratoxigenic *Penicillium nordicum* and *Aspergillus westerdijkiae* [43,44]. Similarly, a medium elaborated with dry-cured ham was inoculated with biocontrol agents and a cellophane sheet was laid over the surface of the agar before the inoculation of ochratoxigenic *P. nordicum* to prevent cross-contamination between the different microorganisms [64]. 

Once the study matrix has been selected, different techniques can be carried out for the proteomic analyses of moulds of interest in food. The most affordable techniques include one- or two-dimensional gel electrophoresis, the latter, 2-DE, being the most common one due to the separation of proteins by two properties, molecular mass and isoelectric point (pI; Figure 1). However, the protein profile obtained from eukaryotic cells, such as moulds, is too complex to be resolved only by 2-DE. In general, the 2-DE works in a limited range of pI, excluding from the resolved part of the gel those more cationic or anionic proteins. The appearance of spots in the gels can provide information about protein identification acting like a map, while the intensity of those spots provides quantitative information about protein levels [65]. For this, the spots should be analysed with image comparison software. This methodology is slow and labour-intensive, which can contribute to a loss of sensitivity, such as those reported when it is used in parallel to other more advanced ones [63,66,67], as discussed later. Thus, this approach usually entails a low efficiency in protein identification and discrimination between batches and even the appearance of human errors. 

Subsequently, the spot identification entails the conversion of the mould proteins individually excised from the 2-DE into peptides by digestion and their analyses by mass spectrometry (MS; Figure 1). Before this, the samples must be purified to remove gel contaminants. The peptides can be analysed using different equipment for mass analysis as an ion trap [67], which cannot offer high resolution; although, it usually achieves enough sequence coverage to identify proteins. All these drawbacks related to protein separation, automatisation, or efficiency in the identification have led to the development of other advanced techniques using cutting-edge technologies, such as time-of-flight (TOF) [68,69], Orbitrap [64,70,71], and Fourier transform ion cyclotron resonance mass spectrometers (FT-ICR-MS), all of them being categorised as high resolution mass spectrometry (HRMS) equipment. The TOF instruments have some advantages with respect to the Orbitrap analyser due to their speed performing full scans, allowing them to match well with ion mobility technologies. Despite this, TOF only achieves mass resolutions between 60,000 and 200,000 full width at half maximum (FWHM), while Orbitrap analysers reach up to 1 million FWHM [72]. However, the time for analysis with the Orbitrap is longer than that used in a TOF, which can lead to lower performance. The FT-ICR-MS has not been used in the study of the mould proteome yet despite its high accuracy achieving resolutions exceeding 2.7 million FWHM [72]. Currently, ion mobility separation techniques are being used in other fields, such as human medicine, for a further separation step in the mass spectrometer improving the measurement’s sensitivity and multiplexing capability [73]. During a trapped ion mobility spectrometry (TIMS) separation event, trapped peptide ions are concentrated and eluted and, together with a TOF, it can analyse multiple targets per accumulation in a short time without compromising sensitivity [73]. This is a promissory tandem-measurement parameter to be coupled to HRMS, as will be discussed later.

## 4. Proteomic Strategies for Improving Food Safety and Quality 

In general, proteomic techniques that allow the monitoring of protein levels can better reveal the metabolic process of moulds and can be used both to detect mould diversity in a microbial community as well as contribute to a better understanding of the mechanisms of action of different antifungal treatments and prevent any risk of mycotoxin contamination or food spoilage. 

### 4.1. Detection and Identification of Moulds

Proteomics for assessing food quality and safety has been applied for a long time with the use of analytical methods, achieving rapid and reliable analysis of food throughout the food chain. One of the strategies to improve food quality and safety related to moulds involves their identification. 

The detection of moulds by proteomic strategies focuses on two main techniques: the traditional and widely used matrix-assisted laser desorption/ionisation time-of-flight mass-spectrometry (MALDI-TOF MS) [74] and the omics approach, metaproteomics, which allows not only the identification of proteins, but also the observation of their PTM [75].

MALDI-TOF MS is based on the acquisition of protein mass spectrum fingerprints from an unknown isolate, which is then identified by comparing its mass spectrum data with those from a reference library [76]. Mass spectrometric peptide/protein profiles of moulds display peaks in the *m/z* region of 1000–20000, where a unique set of biomarker ions may appear facilitating a differentiation of samples at the level of genus, species, or strain [74]. MALDI-TOF MS analysis of subproteomic mass spectra has been shown to be a promising tool for species identification and differentiation in moulds [74,77,78]. Reliable species identification by MALDI-TOF MS has been reported for both food spoilage [79,80] and foodborne moulds [81,82]. Furthermore, its usefulness has also been demonstrated in the identification of moulds in various types of foods, such as ripened cheeses [83,84,85] and asparagus [86].

Metaproteomics is an omics technique able to detect microorganisms in complex microbial communities, such as some foods. Proteins constitute the largest amount of cellular material, and therefore, total per species protein can be quantified to assess the biomass of every member of the microbial community [87]. In addition, these proteins can be assigned to individual species or higher taxa using a protein sequence database and lead to an understanding of the functional roles and interactions of individual members in the community [88]. Metaproteomic provides “snapshots” of microbial populations and can be used to directly study the nature of microbial function in specific environments and states as well as to understand complex substrate–microbiome interactions [89]. Although metaproteomics is a very powerful method, some problems in bioinformatics evaluation impede its large-scale application since this analysis is a key part of this omics methodology. Protein identification requires software and platforms, such as Unipept, MetaLab, ProteoStorm, or Galaxy. Commonly used functional information databases include Cluster of Orthologous Groups (COG), Gene Ontology (GO), and Kyoto Encyclopaedia of Genes and Genomes (KEGG) [89]. In particular, the construction of databases for protein identification, clustering of redundant proteins, and taxonomic and functional identifications pose great challenges [90]. This is one of the main factors why its application to mould detection remains relatively underutilised. 

Metaproteomic techniques used for the detection of microorganisms in foods have been applied mainly in fermented foods to describe not only the microbial composition and succession, but also its role in the process and the relationship of microorganisms with flavour development [91]. Metaproteomics can be used as a tool to optimise food fermentations, for example, by knowing the metabolic pathways, it is possible to choose the starter strains to produce specific metabolites of interest, to know the best nutrients to supplement the medium, and to enhance the performance of the starter or choose intermediate strains and know the appropriate moment to introduce them [75]. Overall, metaproteomics has facilitated the detection of moulds belonging to the *Aspergillus*, *Mucor*, *Rhizopus*, *Penicillium*, and *Geotrichum* genera in various types of soy sauce foods [92,93,94] and fermented beverages [91,95].

### 4.2. Study of Mould Growth and Physiology

Proteomic studies provide a comprehensive vision of the differential protein accumulation during mould growth and the generation of mould secondary metabolites, arising as an important contribution to the identification of new proteins and genes linked to the biosynthesis of mycotoxins (Table 2). These findings contribute to a deeper knowledge of the pathways linked to mycotoxin production and can be very helpful for designing preventive actions to minimise mould spoilage and mycotoxin production in food. The identification of the proteins involved in this metabolite synthesis requires a comparison between the proteome of mycotoxin-producing and non-producing strains, and the proteome of producing strains under conditions of production and non-production of the toxin.

The proteome analysis of *A. flavus* showed that in aflatoxin-producing conditions some aflatoxin biosynthetic enzymes, such as *O*-methyltransferase A (OmtA), AflK/vbs/VERB synthase, ver-1, norA, ver-1, and aflatoxin B1-aldehyde reductase GliO-like, are prevalent in the mycelium, together with proteins from metabolic processes [96]. In this sense, proteins related to aflatoxin biosynthesis, such as AflR, nonribosomal peptide synthetase 10, subunits α and ẞ of fatty acid synthase, sterigmatocystin biosynthesis P450 monooxygenase, polyketide synthase (PksA), noranthrone synthase, noranthrone monooxygenase, NOR reductase, averantin hydrolase, oxidase, esterase, desaturase, and alcohol dehydrogenase, are expressed when mould grows in a favourable substrate, such as corn flour. On the other hand, most of proteins involved in aflatoxin biosynthesis (*O*-methyl sterigmatocystin oxidoreductase, sterigmatocystin 8-*O*-methyltransferase, p450 monooxygenase AflN, versicolorin B desaturase, averufin oxidase A, averantin hydroxylase, and noranthrone synthase), as well as in carbohydrate metabolism, cell wall biogenesis, mitogen-activated protein kinase (MAPK) signalling pathways, heat shock proteins, autophagy, and dicer-like proteins, are yet expressed at the germinating conidial stage. These data suggest that the MAPK signalling pathway could be crucial in cell wall modulation and secondary metabolite synthesis, and that the biosynthesis of aflatoxins could start at early germination stages with favourable conditions [97].

In the proteome of two strains of *A. carbonarius* differing in their ochratoxin A-producing potential, nine proteins (seven increased and two reduced in quantity) were detected as potentially involved in several biological functions, such as regulation, amino acid metabolism, oxidative stress, and sporulation [14]. Among them, a protein homologous to CipC showed the highest relative abundance in the ochratoxin A-producing strain. Although the function of this protein is still unknown, it was concluded that it is probably involved in ochratoxin A biosynthesis [14].

The composition of the substrate and the environmental conditions have a major impact on mould physiology, and consequently, these changes should also be apparent in the proteome. Thus, proteomics has been applied to explore the impact of different external factors, such as water activity (a_w_), temperature, pH, nutrient substrate, salt content, or light on several foodborne moulds and mycotoxin production. 

The response to different a_w_ in *A. flavus* resulted in variations in the relative amount of 837 proteins; 403 at higher abundance at 0.99 a_w_ and 434 more abundant at 0.93 a_w_ [98]. Osmotic stress-related proteins, Sln1 and Glo1, belonging to the Hog1 pathway showed higher levels at 0.99 a_w_. These results are consistent with the fact that *A. flavus* grows better under high a_w_ conditions [98]. The secretion of extracellular hydrolases increased as a_w_ rose, suggesting that they may play a critical role in the induction of aflatoxin biosynthesis. Furthermore, the export protein KapK may downregulate aflatoxin biosynthesis with the translocation of NirA, a specific transcription factor in the nitrate assimilation pathway. In addition, the abundance of 11 proteins directly related to aflatoxin biosynthesis (aflE, aflF, aflH, aflJ, aflK, aflM, aflO, aflP, aflQ, aflY, and aflYa) were higher at 0.99 a_w_, and just one (aflYc) was more abundant at 0.93 a_w_ [98]_._ The *aflE* and *aflF* genes encoding ketoreductases that convert norsolorinic acid to averantin in the aflatoxin synthesis pathway were expressed only in aflatoxin supportive conditions (0.99 a_w_) [98]. These data are valuable for understanding the impact of water stress on aflatoxin production and for the design of preventive measures for its control in foods [98].

In the proteome of *A. flavus* growing at temperatures of 28 °C and 37 °C, using the iTRAQ labelling, 664 proteins were found in different relative abundance, especially belonging to translation-related pathways, metabolic pathways, and the biosynthesis of secondary metabolites [55]. The growth, but not the production of aflatoxins by *A. flavus*, is favoured at 37 °C, while the opposite occurs at 28 °C. In this sense, 12 aflatoxin biosynthesis-related proteins (aflE, aflW, aflC, aflD, aflO, aflP, aflK, aflM, aflY, aflJ, aflS, and aflH) showed a higher abundance at 28 °C than at 37 °C [55]. By SILAC tagging, 31 proteins were found in higher amounts at 37 °C (including AflM and AflP) and 18 were more abundant at 28 °C (including AflD, AflE, AflH, and AflO). The shift in the expression of the aflatoxin pathway enzymes is closely related to the strong repression of both aflatoxin biosynthesis and transcription of the aflatoxin pathway genes observed at 37 °C [99]. The pathway-specific regulatory *afl*R gene, required for the activation of most aflatoxin pathway genes, was upregulated at 28 °C [55], but the aflR protein was not detected in the proteomic profiles of *A. flavus* at either 28 °C or 37 °C [55,99]. Likewise, the aflR protein was not detected in *A. flavus* grown under different a_w_ conditions, regardless of whether or not they favour aflatoxin production [98]. These results lead to conclude that there is a low correlation between proteome and transcriptome data, suggesting that post-transcriptional gene regulation affects distinct biological pathways and secondary metabolite gene clusters [55].

In a transcriptome and proteome analysis conducted to clarify the mechanisms explaining the higher production of aflatoxin B_1_ by *A. flavus* in maize and rice broth than in peanut broth, fewer differences in the gene expression and protein abundances were observed between the maize and rice substrates than between the above substrates and peanut [58]. Most of the proteins with different amounts are involved in metabolic process, cellular process, catalytic activity, binding, cell, and cell part, but the limited variations suggest that the growth and metabolism of *A. flavus* in these substrates are similar, mainly in rice and maize. The expressions of genes linked to the early phase of aflatoxin biosynthesis (*aflA*, *aflB*, and *aflC*) and the *accA* gene were significantly increased in maize and rice substrates. Genes related to carbon metabolism were upregulated in maize broth, while those involved in acetyl-CoA accumulation and consumption were up- and downregulated, respectively. Several genes involved in the aflatoxin biosynthesis regulation, namely *veA*, *ppoB*, *snf1*, and G protein-coupled receptor (GPCR) genes, were differentially expressed in the three substrates, indicating that they may also be involved in the sugar signal sensing, transfer, and regulation. Notably, correlation analyses of the transcriptome and proteome showed that the trehalose metabolism genes, the aldehyde dehydrogenase gene, and the tryptophan synthase gene are important in the regulation of aflatoxin yield in different substrates [58]. A low correlation of transcriptome and proteome data was obtained, similarly to the abovementioned studies regarding the effect of a_w_ and temperature on the production of aflatoxin by *A. flavus* [55]. This finding could be due to the insufficient number of recovered proteins, the different synthesis and turnover rate of proteins and mRNAs in various cell stages, and the post-transcriptional or post-translational modifications [58].

**Table 2 ijms-24-04709-t002:** Proteomic studies on mechanisms of mould growth and mycotoxin production.

Mould	Purpose	Proteomic Methodology Used	Main Proteomic Findings	References
*Aspergillus flavus*	Proteome profile of mycelium	2-DE and mass spectrometry	Proteins from cellular metabolic process, and AFs ^a^ biosynthesis are prevalent.	[96]
*A. flavus*	Proteome during germination of conidia	LC-mass spectrometry	AFs biosynthesis, carbohydrate metabolism, cell wall biogenesis, MAPK pathways, heat shock, autophagy, and dicer-like proteins are expressed at germinating conidia.	[97]
*A. flavus*	Proteome on corn flour	LC-mass spectrometry	↑ AFs biosynthesis proteins.	[100]
*A. flavus*	Influence of water activity (a_w;_ 0.99 vs. 0.93)	LC-mass spectrometry	↑ Proteins related to osmotic stress and Afs biosynthesis at 0.99 a_w_.	[98]
*A. flavus*	Influence of temperature (28 °C vs. 37 °C)	LC-mass spectrometry	Changes on proteins belonging to translation pathways, metabolic pathways, and secondary metabolites biosynthesis.	[55]
*A. flavus*	Influence of temperature (28 °C vs. 37 °C)	LC-mass spectrometry	Changes in abundance of 49 proteins, including AFs pathway, leading to repression of AFs biosynthesis at 37 °C.	[99]
*A. flavus*	Influence of substrate (maize/rice vs. peanut) on AFs production	LC-mass spectrometry	Few differences in protein abundances between crops. Low correlation between transcriptome and proteome data.	[58]
*Aspergillus carbonarius*	Proteomic profile according to OTA ^b^-producing potential	2-DE and mass spectrometry	CipC protein likely involved in OTA biosynthesis.	[14]
*Aspergillus ochraceus*	Influence of NaCl content (20 g/L vs. 70 g/L) on OTA production	LC-mass spectrometry	Changes in proteins involved in nutrient uptake, membrane integrity, cell cycle, energy metabolism, redox homeostasis, protein synthesis, autophagy, and secondary metabolism. ↑ OTA with 20 g/L NaCl.	[62]
*Aspergillus niger*	Influence of the substrate and culture conditions	2-DE and mass spectrometry	Extracellular proteome mainly influenced by the carbon source and intracellular proteome by the environmental conditions.	[101]
*A. niger*	Influence of addition of lactate on mycotoxin production	2-DE and mass spectrometry	Changes in proteins related to acetyl-CoA or NADPH, pentose phosphate pathway, pyruvate metabolism, tricarboxylic acid cycle, ammonium assimilation, fatty acid biosynthesis, and oxidative stress protection. ↑ Fumonisin B_2_; ↓ OTA.	[102]
*Penicillium digitatum*	Effect of limonene on the growth	LC-mass spectrometry	Changes in proteins related to energy metabolism and antioxidant defence.	[103]
*Penicillium expansum*	Interaction with apple	2-DE and mass spectrometry	↑ Proteins related to pathogenesis, secondary metabolism, and patulin biosynthesis.	[7]
	Interaction with apple	LC-mass spectrometry	↑ Proteins related to growth, stress tolerance, and virulence.	[104]
*Penicillium verrucosum*	Effect of short wavelength light	2-DE and mass spectrometry	Changes in proteins involved in stress response and metabolic processes. ↑ Citrinin; ↓ OTA.	[9]
*Fusarium proliferatum*	Influence of pH (5 vs. 10) on fumonisin production	2-DE and mass spectrometry	↑ Proteins involved in fumonisin backbone modification and ↑ fumonisin at pH 10.	[105]
*Neosartorya pseudofischeri*	Resistance to heat (93 °C)	LC-mass spectrometry	↓ Proteins involved in carbon metabolism, heat stress responses, ROS ^c^ elimination, and translation pathways.	[106]

^a^ AFs: aflatoxins; ^b^ OTA: ochratoxin A; ^c^ ROS: reactive oxygen species.

The addition of salt in the processing of a variety of foods, such as dry-cured meat and dairy products, favours the growth of both beneficial and toxigenic moulds on their surface. Particularly in meat products, the proliferation of ochratoxin A-producing moulds is of special concern. Therefore, from a food safety perspective, the study of the influence of the salt added to meat products on the growth of these moulds and the production of toxins is of great interest. The addition of 20 g/L NaCl in a culture medium induced the spore production of *Aspergillus ochraceus*, while 70 g/L NaCl repressed it [107]. Comparative proteomics analysis of *A. ochraceus* growing with 20 or 70 g/L NaCl revealed significant changes in the abundance of proteins involved in nutrient uptake, cell membrane integrity, cell cycle, energy metabolism, intracellular redox homeostasis, protein synthesis and processing, autophagy, and secondary metabolism. The latter activity, including ochratoxin A production, was stimulated by the addition of 20 g/L NaCl, with an increase of non-ribosomal peptide synthetases (NRPS), and repressed by 70 g/L NaCl. At the highest concentration, an increased extracellular hydrolase production was observed, probably for the adaptation to nutrient starvation due to a decrease in energy metabolism. A higher concentration of reactive oxygen species (ROS) was also detected, which was harmful for protein synthesis and even triggered autophagy [107]. 

The extracellular proteome of *Aspergillus niger* differed considerably depending on the carbon substrate xylose or maltose [101]. When the medium was supplemented with xylose, a variety of plant cell wall degrading enzymes were identified with xylanase B and ferulic acid esterase as the most abundant ones. In cultures with maltose, high levels of catalases were found and glucoamylase was the most abundant protein. However, the intracellular proteome was not significantly changed. Interestingly, other culture conditions, such as pH control, aeration, stirring, or shaking, strongly influenced the abundance of glycolytic and tricarboxylic acid (TCA) cycle enzymes, flavohemoglobin, CipC protein, superoxide dismutase, NADPH-dependent aldehyde reductase, ER-resident chaperones, and foldases [101] in the intercellular proteome. On the other hand, the addition of lactate in a medium containing starch and nitrate provokes an increase in the production of fumonisin B_2_, but not of ochratoxin A by *A. niger* [102]. The proteome of *A. niger* was affected, mainly in the abundance of proteins related to the intracellular level of acetyl-CoA or NADPH, such as enzymes in the pentose phosphate pathway, pyruvate metabolism, the TCA cycle, ammonium assimilation, fatty acid biosynthesis, and oxidative stress protection. These data support the hypothesis that fumonisin production by *A. niger* is regulated by acetyl-CoA [102].

On the other hand, some compounds from foods can stimulate the germination of undesirable moulds. For example, limonene, a dominant volatile constituent in oil glands of most citrus, promotes spore germination, germ tube elongation, and mycelial growth of the citrus pathogen *P. digitatum*. Limonene alters the abundance of 340 proteins in *P. digitatum* including proteins related to energy metabolism and antioxidant proteins, such as glutathione S-transferases, superoxide dismutase, and catalases. Limonene thus induces the growth of *P. digitatum*, probably through the regulation of energy metabolism and ROS homeostasis [103].

*P. expansum* commonly causes blue mould rot and postharvest decay in apples, pears, and other pome fruits, and is the main producer of patulin. Proteomics has been used for studying the molecular mechanism involved in the interaction of this mould with apple fruit. In an apple substrate, 28 proteins forming *P. expansum* were found in higher relative abundance. These proteins were mainly associated with pathogenesis, such as glyceraldehyde-3-phosphate dehydrogenase, catalase, and peptidase, and with secondary metabolism and patulin biosynthesis regulation, for instance, glucose dehydrogenase and FAD-binding monooxygenase [7]. These changes in the proteome might be responsible for the observed medium acidification and patulin production [7]. In the proteome of *P. expansum* growing on apple juice, up to 148 proteins were found in a high quantity, including cell-wall degrading enzymes and peptidases/proteases, especially a serine carboxypeptidase (PeSCP) required for conidiation, germination, fungal growth and morphology, tolerance to environmental stresses, extracellular carboxypeptidase activity, and fungal virulence [104].

The influence of pH on the production of fumonisin by *Fusarium proliferatum* has been explored by proteomic analysis. The increase of fumonisin production at pH 10 was related to the higher quantity proteins, such as polyketide synthase, cytochrome P450, S-adenosylmethionine synthase, and *O*-methyltransferase, involved in the modification of the fumonisin backbone. In contrast, at pH 5, the higher abundance of L-amino-acid oxidase, isocitrate dehydrogenase, and citrate lyase was linked to the inhibition of the condensation of the fumonisin backbone and the concurring decrease of the mycotoxin production [105].

The exposure to light of short wavelengths induces oxidative stress in *Penicillium verrucosum* together with a marked decrease in the synthesis of ochratoxin A and a significant increase in the production of citrinin. Through a proteomic analysis combining two-dimensional SDS-PAGE with HPLC-ESI-TOF-MS/MS, 56 significantly differential proteins between cultures grown in light versus dark were detected. Most of them are presumably involved in the stress response, such as antioxidant proteins or heat shock proteins, and in general, metabolic processes, for example, glycolysis or ATP supply [9].

*Neosartorya pseudofischeri* is a heat-resistant fungus and can contaminate several juices. The cellular process of heat-resistance has been studied in ascospores subjected to heat treatment at 93 °C for 0, 1, or 8 min. A total of 150 proteins significantly altered in abundance were identified, of which, 126 showed decreased abundance after heat treatment mainly involved in the central carbon metabolism, heat stress responses, reactive oxygen intermediates elimination, and translation events. These proteins are potential targets to evaluate the efficiency of thermal treatment for processed food products [106].

### 4.3. Mode of Action of Antifungal Agents against Foodborne Moulds

Several antifungal agents have been proposed to control the growth of undesirable moulds and mycotoxin accumulation on foods, including microorganisms and chemical compounds. To study the efficacy of these antifungal agents, deciphering both their mechanisms of action and their cellular targets in the moulds of interest is a crucial issue. A proper understanding of the target can provide valuable information on the spectrum of activity of the control agents and the possible sensitivity of the different toxigenic moulds. Moreover, information on possible modes of resistance can be obtained, as well as to guide designing strategies using combinations of different control agents that affect distinct targets. Potential side effects, such as the generation of unwanted by-products of treatment, such as mycotoxins or other undesirable secondary metabolites, can also be elucidated. 

Proteomic studies have provided valuable knowledge about the systems disturbed in response to antifungal agents and they have been applied to characterise the behaviour of both resistant and susceptible moulds, allowing for the recognition of mechanisms of resistance as well as the identification of promising susceptible targets (Table 3). Overall, these methods underline the range of tools available to provide a global overview of the molecular targets and biological pathways impacted by antifungal agents. These comprehensive perspectives can support the further targeting of complementary techniques, such as biochemical analysis, targeted gene disruption, or metabolite profiling.

Proteomic analyses have been conducted to clarify the mode of action of antifungal agents against several toxigenic moulds and mycotoxin production, for example *P. nordicum*, *Aspergillus westerdijkiae*, *A. flavus*, *A. carbonarius*, *P. digitatum, P. expansum, P. italicum,* and *Fusarium oxysporum*. Concretely, *P. nordicum* and *A. westerdijkiae* have undergone many studies employing a variety of biocontrol agents as they have been described as the main producers of ochratoxin A in meat products. *A. flavus* is the subject of numerous studies for its control since it is the main producer of the high toxic and carcinogenic aflatoxins. 

*Penicillium chrysogenum* and *Debaryomyces hansenii* repressed ochratoxin A production by *P. nordicum* in a dry-cured ham-based medium, likely by nutritional competition. According to proteomic data, both agents inhibited *P. nordicum* through cell wall integrity (CWI) impairment, and they hamper the secondary metabolism, including ochratoxin A synthesis, lowering the levels of MAPK, and the carbon catabolite repression (CCR) pathway [64]. Rosemary essential oil decreased the abundance of proteins involved in the polyketide synthase enoylreductase (PKS ER) domain in *P. nordicum*, which would explain the ochratoxin A reduction. The combination of rosemary leaves with *D. hansenii* lowered the abundance of proteins linked to the CWI and purine pathway [44]. Rosemary essential oil decreased the abundance of proteins involved in the polyketide synthase enoylreductase domain in *P. nordicum*, which would explain the ochratoxin A reduction, and the mixture of rosemary leaves with *D. hansenii* reduced the abundance of proteins related to the CWI and purine pathway [44].

*D. hansenii* singly or in combination with rosemary or its essential oil causes a large reduction in the production of ochratoxin A by *A. westerdijkiae* lowering the abundance of proteins involved in ochratoxin A production, such as PKS ER and NRPS, and in the CWI pathway [43]. On the other hand, the combination of rosemary leaves and its essential oil decreases the ochratoxin A production disturbing the abundance of proteins from the PKS ER domain and CWI pathway of *A. westerdijkiae* [70].

Volatile compounds generated by yeasts have demonstrated inhibitory effects against toxigenic moulds [108,109]. The volatilome of *Candida intermedia* reduces the growth, sporulation, and ochratoxin A biosynthesis by *Aspergillus carbonarius* [109]. Both the volatilome of *C. intermedia* and its major component 2-phenylethanol affected a variety of metabolic targets, the most concerned routes being the central metabolism, the energy production, and the stress response. Volatilome has a stronger effect on protein biosynthesis. Although 2-phenylethanol impacts some metabolic traits, other unidentified volatile components may involve a plurality of metabolic targets that may result in a higher effectiveness of the volatilome [46].

Several compounds have been considered for the control of *A. flavus*, such as the antifungal protein PgAFP, the quercetin, and the *Perilla frutescens* essential oil (PEO). PgAFP, an antifungal protein secreted by a strain of *P. chrysogenum* [110], has been studied as a biocontrol agent against toxigenic moulds on dry-cured foods [111]. PgAFP provoked apoptosis and necrosis in *A. flavus* hyphae with the reduction of energy metabolism, alteration of CWI, and increase of ROS. Label-free mass spectrometry-based proteomics (Figure 2) showed changes in the proteome of *A. flavus*, with higher glutathione and heat shock proteins concentrations, and lower relative quantity of Rho1 and the β subunit of G-protein [63]. However, PgAFP did not alter the metabolic capability, chitin deposition, or hyphal viability of *A. flavus* grown in cheese due to the calcium content. A total of 125 proteins were increased in the presence of calcium, including oxidative stress-related proteins, whereas 70 proteins were found at lower abundance, mainly involved in metabolic pathways and the biosynthesis of secondary metabolites. The resistance conferred by calcium to *A. flavus* appears to be mediated by calcineurin, G-protein, and g-glutamyltranspeptidase, which combat oxidative stress and impede apoptosis [67]. On the other hand, a strain of *Penicillium polonicum* is natively resistant to PgAFP by increasing chitin content of its cell wall. Proteome changes allow for the attribution of this resistance to a higher abundance of glucosamine-6-phosphate N-acetyltransferase and Rho GTPase Rho1 that would lead to the increased chitin deposition via CWI signalling pathway [66]. Therefore, proteomics has shed light on the mode of action of the antifungal protein PgAFP and some native or acquired mechanisms of mould resistance. This information is useful to design strategies to improve the PgAFP activity against toxigenic moulds in foods.

Quercetin induces various oxidative stress response proteins, but suppresses the MAPK pathway and the expression of several enzymes involved in the aflatoxin biosynthesis, such as AflR, acetyl CoA synthetase, noranthrone synthase, noranthrone monooxygenase, NOR reductase, averantin hydrolase, sterigmatocystin biosynthesis polyketide synthase, and *O*-methyl transferase A [100].

A comparative global proteomic analysis of *A. flavus* using the TMT labelling method revealed that PEO inhibits the growth of *A. flavus* by blocking the antioxidative defence, reducing the expression of superoxide dismutase and catalase associated with the elimination of ROS. Moreover, several proteins, such as ATP-dependent 6-phosphate fructokinase, triosephosphate isomerase, glyceraldehyde 3-phosphate glyceraldehyde dehydrogenase, and phosphoglycerate, were found in lower relative quantities, repressing the glycolysis pathway leading to the disturbance of energy metabolism, which cannot be overcome by *A. flavus*, even though additional energy-producing pathways, for instance, fatty acid degradation, amino acid metabolism, pyruvate metabolism, and glyoxylic acid metabolism, were activated [2].

The essential oil citral, comprised of a mixture of terpenoids, geranial and neral, inhibits the *A. ochraceus* growth and ochratoxin A production by accumulation of ROS, resulting in the damage of mould cell membranes and cell walls. The treatment with subinhibitory concentrations of citral altered the amount of 218 proteins of *A. ochraceus* proteome studied by iTRAQ, perturbating proteins involved in the fungal growth and development, nutrient intake, and energy metabolism. Conversely, proteins associated with cell wall maintenance, membrane integrity, antioxidative defence, and secondary metabolism were increased. Nevertheless, the answer proved to be insufficient to overcome the stress resulting from citral-mediated ROS accumulation and repression of cell growth, resulting in a lower accumulation of ochratoxin A [62].

The yeast *Pichia caribbica* has been proposed as a biocontrol agent against *P. expansum* in apples, and its effects are enhanced by vitamin C, by increasing the abundance of proteins related to the glucose metabolism, such as glyceraldehyde-3-phosphate dehydrogenase and alcohol dehydrogenase. These changes allowed the growth increase of *P. caribbica* and then enhanced its inhibitory effect over *P. expansum* [45].

**Table 3 ijms-24-04709-t003:** Changes induced by control agents in the proteome of undesirable foodborne moulds.

Antifungal Agent	Target Mould	Proteomic Method Used	Main Proteomic Findings	References
*Penicillium chrysogenum*	*Penicillium nordicum*	2-DE and LC-mass spectrometry	↓ CWI ^a^ and secondary metabolites biosynthesis proteins.	[64]
*Debaryomyces hansenii*	*P. nordicum*	2-DE and LC-mass spectrometry	↓ CWI and secondary metabolites biosynthesis proteins.	[64]
*D. hansenii*	*Aspergillus westerdijkiae*	LC-mass spectrometry	↓ CWI and OTA ^b^ biosynthesis proteins.	[43]
*Candida intermedia*	*Aspergillus carbonarius*	LC-mass spectrometry	↓ Central metabolism, energy production, and stress-response proteins.	[46]
*Rosmarinus officinalis*	*P. nordicum*	LC-mass spectrometry	↓ OTA biosynthesis.	[44]
*R. officinalis*	*A. westerdijkiae*	LC-mass spectrometry	↓ CWI and OTA biosynthesis proteins.	[70]
Protein PgAFP	*Aspergillus flavus*	2-DE and LC-mass spectrometry	↑ Stress-response proteins. ↓ CWI proteins.	[63]
Protein PgAFP	*A. flavus* cultured on calcium-enriched substrate	2-DE and LC-mass spectrometry	↑ Oxidative stress-response proteins and secondary metabolites biosynthesis.	[67]
Protein PgAFP	*Penicillium polonicum*	2-DE and LC-mass spectrometry	↑ CWI proteins.	[66]
Quercetin	*A. flavus*	LC-mass spectrometry	↑ Oxidative stress-response proteins. ↓ MAPK pathway and AFs ^c^ biosynthesis proteins.	[100]
*Perilla frutescens*	*A. flavus*	LC-mass spectrometry	↓ Antioxidative and glycolysis pathway proteins. ↑ Fatty acid, amino acid, pyruvate, and glyoxylic acid metabolism proteins.	[2]
α-sarcin and beetin 27 proteins	*Penicillium digitatum*	LC-mass spectrometry	Changes on cell wall-degrading, stress response, antioxidant, and detoxification mechanisms and metabolic processes proteins.	[112]
2-methoxy-1,4-naphthoquinone	*Penicillium italicum*	LC-mass spectrometry	Changes on energy generation, NADPH supply, oxidative stress, and pentose phosphate pathway proteins.	[113]
Pinocembrin	*P. italicum*	LC-mass spectrometry	Changes on mitochondrial respiratory chain complexes I and V. ↑ PCD ^d^-related proteins.	[114]
Tea tree oil	*Botrytis cinerea*	LC-mass spectrometry	↓ TCA ^e^, pyruvate, and amino acid metabolism, and membrane-related pathways. ↑ Sphingolipid metabolism.	[48]
Chitosan	*Fusarium oxysporum*	2-DE and mass spectrometry	↓ Virulence proteins and ROS ^f^-degrading enzymes.	[115]
Chitosan	Antifungal activity of *Rhodotorula mucilaginosa*	LC-mass spectrometry	↑ Energy metabolism, antioxidant and abiotic stress response, and degradation of pathogen cell.	[71]
Vitamin C	Antifungal activity of *Pichia caribbica*	2-DE and mass spectrometry	↑ Glucose metabolism enzymes.	[45]
*Wickerhamomyces anomalus*	Resistance of pear to fungi	2-DE and mass spectrometry	↑ Resistance-related proteins	[116]

^a^ CWI: cell wall integrity pathway; ^b^ OTA: ochratoxin A; ^c^ AFs: aflatoxins; ^d^ PCD: programmed cell death; ^e^ TCA: tricarboxylic acid cycle; ^f^ ROS: reactive oxygen species.

*P. digitatum* is responsible for the postharvest decay of citrus. A proteomic approach based on isobaric labelling and a nanoLC tandem mass spectrometry was used to explore changes in the mould as a response to treatments with the antifungal proteins α-sarcin and beetin 27, inhibitors of protein synthesis compared with those triggered by the chemical fungicide thiabendazole. Results showed differentially expressed proteins between treatments, including mainly cell wall-degrading enzymes, stress response proteins, antioxidant and detoxification mechanisms, and metabolic processes, such as thiamine biosynthesis, suggesting the existence of peculiar responses to each treatment [112]. 

*P. italicum* is considered the principal cause of blue mould of citrus. The natural 2-methoxy-1,4-naphthoquinone (MNQ), isolated from the traditional Chinese medicinal plant *Impatiens balsamina*, had an anti-*P. italicum* effect. Analysing the proteome under different MNQ treatments, 129 proteins with differential quantity were identified, mainly related to energy generation (mitochondrial carrier protein, glycoside hydrolase, acyl-CoA dehydrogenase, and ribulose-phosphate 3-epimerase), NADPH supply (enolase and pyruvate carboxylase), oxidative stress (catalase and glutathione synthetase), and pentose phosphate pathway (ribulose-phosphate 3-epimerase and xylulose 5-phosphate). Thus, the inhibition of *P. italicum* by MNQ may be attributed to the disruption of the metabolic processes, especially the energy metabolism and the stimulus response [113]. Pinocembrin is a flavonoid from propolis active against *P. italicum*. The treatment provokes in the proteome of *P. italicum* (studied by iTRAQ) the alteration in the relative abundance of proteins from the mitochondrial respiratory chain (MRC) complexes I and V and an increasing of proteins related to the programmed cell death, resulting in a ROS accumulation and ATP depletion, which may lead to the cell death through apoptosis, autophagy, and necrosis mechanism [114].

Chitosan is a natural biocompatible, biodegradable, and non-toxic polysaccharide derived from chitin obtained from crustacean shells. Chitosan has been used both to inhibit pathogenic moulds, such as *F. oxysporum*, and to enhance the antifungal activity of some biocontrol agents, such as *Rhodotorula mucilaginosa*. 

*F. oxysporum* f. sp. *cucumerinum* that causes yield losses in cucumber plants is sensitive to chitosan that restricts plant disease severity. A proteomic approach using 2-DE coupled with LC-MS/MS analysis identified 62 differentially abundant chitosan-responsive proteins, most with proteolysis and hydrolase activity involved in metabolism and defence. Chitosan-treated *F. oxysporum* showed a lower abundance of proteins responsible for virulence, such as plant cell wall-degrading enzymes, structural and functional protein and DNA biosynthesis, and transporter proteins. Moreover, a decrease of the ROS-degrading enzymes glutathione peroxidase and catalase-peroxidase may result in ROS accumulation that can induce apoptosis, reducing mould virulence [115].

The efficacy of *R. mucilaginosa* against the grey mould *B. cinerea*, which causes a postharvest disease of fruits and vegetables, can be enhanced by previously culturing the yeast in a medium containing chitosan. Chitosan triggered in *R. mucilaginosa* the higher quantity of proteins involved in the growth and reproduction, energy metabolism, antioxidant response, response to abiotic stress, and degradation of the pathogen cell. These changes can increase the growth rate of *R. mucilaginosa* and improve its capability to withstand and survive diverse abiotic stresses, allowing it to better compete for nutrients and space against *B. cinerea* [71].

The spore germination and mycelial growth of *B. cinerea* is also inhibited by TTO, which alters the relative abundance of 85 proteins identified by label-free proteomic. The analysed data suggests that the TTO inhibits the TCA cycle, pyruvate metabolism, amino acid metabolism, and membrane-related pathways in mitochondria, and promotes sphingolipid metabolism, which may accelerate cell death in *B. cinerea* [48]. 

The yeast *Wickerhamomyces anomalus* significantly reduces the natural decay of pear fruit. The proteome of pear fruit, analysed using 2-DE and MALDI-TOF/TOF, indicated that *W. anomalus* induces the accumulation of resistance-related proteins, such as PR family proteins, chitinase, and β-1,3-glucanase, which can inhibit the infection of the moulds whose cell walls contain β-1,3-glucan or chitin [116].

Therefore, multiple approaches through proteomic tools have substantially contributed to the unravelling of those mechanisms beyond the biocontrol agents’ effect on foodborne moulds. This information has served to achieve a better understanding of the moulds’ cellular and pathway targets to improve their control. However, there is a wide window of proteomic applications to be introduced in the flowchart of the foodborne moulds analyses that clearly surpass the currently applied ones, to further maximise the results obtained in line with those already achieved.

## 5. Emerging Proteomics to Be Exploited in Foodstuffs 

As discussed before (Section 3), different major advantages have been pointed out in relation to HRMS when compared with 2-DE. Nevertheless, different massive acquisition methods within HRMS are available: data-dependent acquisition (DDA), based on the isolation and fragmentation of the “n” most intense signal peptides, and data-independent acquisition (DIA), which fragments all peptides, disregarding their intensities. Both acquisition methods entail advantages and drawbacks [117].

The main advantage of DDA is the ease of processing after sample HRMS analysis in identifying peptides and proteins. The available software base their procedures, extremely simplifying them, on peptide-spectrum matches, and the number of protein/peptide quantifications corresponds to those having peak intensities [118,119]. A FASTA file containing all the proteins from the target mould is required for these peptide-spectrum matches, usually available if the microorganism genome exists. The relative affordability of this data processing is probably the main reason why most of the reported mould proteomics linked to foodstuff using HRMS, gathered in this review, have used this approach. However, this is not exempt from disadvantages, such as the different magnitude order between the most and the least abundant proteins in a complex matrix, which means the impossibility of the detection of many proteins [117]. This is translated into numerous proteins, probably relevant ones for the mould physiology in foodstuffs, being hidden to the researchers’ eyes when their abundances are below the lowest threshold of the dynamic range.

DIA overcomes DDA concerning the dynamic range coverage, since it relies not on precursors selected individually, but on set systematic windows of precursors, and the fragmentation of all peptide ions contained in these windows [119], oversampling, in comparison to DDA [120]. However, the main disadvantage linked to this approach, although recently overcome as discussed below, has been the requirement of a library building. This is time- and cost-consuming, only affordable for robust and powerful research groups. Nevertheless, different algorithms have been recently published/released, allowing the analyses of DIA raw data without the necessity of the library building, assisted by artificial intelligence (AI). Among the open-source tools, MaxDIA, developed by Max Plank and based on machine learning, has been postulated as an accurate alternative to DIA analyses without requiring a library building [121]. Another interesting open-source tool is DIA-NN [122], which is included in different commercial software [123] and provides the possibility of library-free DIA analyses. Furthermore, DIA-Umpire can perform this task; although the library-free mode has not achieved relevant results. Among them, DIA-NN could be considered the most robust tool for library-free DIA analyses [119]. Looking at commercial tools, Spectronaut and ScaffoldDIA are able to successfully fulfil these [119,120]. Additionally, some of them integrate a new and promising measurement parameter, yet unexploited in fungal proteomics in foodstuff, and considered a fourth dimension in MS, namely ion mobility.

Beyond instrumental techniques based on HRMS, as well as software assisted by AI, to analyse the complexity of fungal proteome in terms of depth and coverage, PTM are still a somewhat unexplored field in food mycology. These comprise mechanisms to enhance the diversity of protein species and functions involved in a wide range of cellular processes [124]. In other far away scientific fields, their study has outstandingly contributed to the scientific advance and development of state-of-the-art technologies, mainly in cancer research [125,126]. These are slowly being implemented in other lower cutting-edge fields, such as our field. So far, the vast majority of the studies involving moulds not related to foodstuff that have evaluated PTMs, relied on phosphorylation [10,127,128], and they were mainly focused on mould–host interactions [10,12]. 

Although PTMs comprise a huge variety of possible variables, some of them offer new perspectives from the physiological view. This is the case of the phosphorylation on serine, threonine, or tyrosine residues as PTM. This reveals the ability to regulate signalling metabolic pathways, kinase cascade activation, membrane transport, gene transcription, and motor mechanisms [129], which otherwise would remain hidden. Again, this PTM measurement has not been extensively exploited in foodborne moulds. However, it has released critical information about the signalling of key pathways, complementary to the relative protein quantification, in non-foodborne moulds, such as *A. fumigatus* or *Aspergillus nidulans* [130,131,132]. This approach would allow for the extraction of alternative information from moulds grown in foods or food systems, in relation to the deciphering of mycotoxin metabolic pathways and mechanisms of action of different antifungal strategies. 

Regarding the PTMs analyses, the aforementioned ion mobility, a new and promissory parameter, is gaining interest for their analyses. It is based on the size/shape features of the ion when it has undergone a gas flow and an electric field, working both in opposite directions. The new dimension that ion mobility offers achieves the distinguishing of signals from peptides that would otherwise be co-fragmented, thus obtaining cleaner spectra [123]. Therefore, this new spectrometric feature allows for a better discrimination between peptides with PTMs when more than a similar residue is susceptible for modification in a given peptide, since this peptide size would be similar, but not their shapes. To the best of our knowledge, this fourth dimension has not been applied to foodborne moulds and it would be of utmost interest to decipher the PTMs, which currently means hidden signalling pathways. 

## 6. Conclusions and Future Trends

An overview of the recent advancements of proteomics applied to foodborne moulds, as well as the potential of approaches based on this high-throughput technology not used yet for such moulds, have been given. Details about the preparation of samples and the techniques applied to evaluate the mould proteins for their identification and the characterisation of the mechanism of action involved in their negative effect on the foodstuffs have been discussed. Metaproteomics seems to be the most powerful method for mould identification despite the current problems related to bioinformatics tools. Proteomics can reveal the molecular mechanisms critical for mould adaptation to their ecological niche. Furthermore, it allows for the understanding of how different external factors, such as environmental conditions, as well as the presence of other microorganisms, may influence the mould development and mycotoxin production. Different HRMS tools have been useful for evaluating the proteome of foodborne moulds; although many authors have combined them with 2-DE despite their disadvantages in comparison with the whole proteomic analysis performed by the first ones. To overcome some of the limitations of the high-throughput technology proteomics applied to foodborne moulds, those used in other scientific fields could be valuable. Concretely, these analyses would be greatly benefited by using library-free DIA analyses, the implementation of ion mobility, and PTMs analysis, as isolated approaches and by combining them. All these are still available, and assuming their economic cost, comprise a various portfolio of improvements to be gradually implemented in this field. Research effort is thus required to address challenges related to the elucidation of key mechanisms of action of foodborne moulds. This knowledge is crucial for the future development of strategies to avoid the presence of unwanted moulds in foodstuffs.

## Figures and Tables

**Figure 1 ijms-24-04709-f001:**
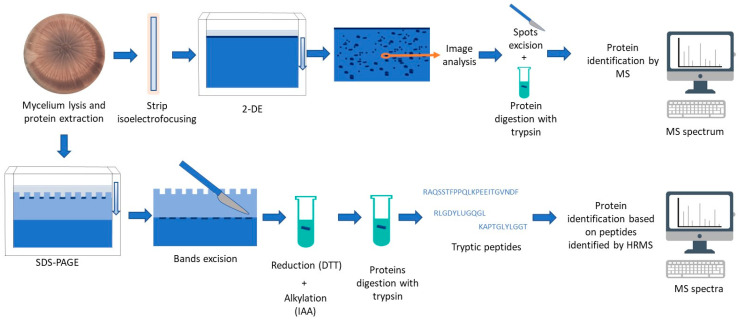
Workflow of proteomics analyses by two-dimensional gel electrophoresis (2-DE), upper line, and sodium dodecyl-sulphate polyacrylamide gel electrophoresis (SDS-PAGE) previous to high resolution mass spectrometry (HRMS) analyses, lower line. DTT: dithiothreitol; IAA: iodoacetoamide-mediated; MS: mass spectrometry.

**Figure 2 ijms-24-04709-f002:**
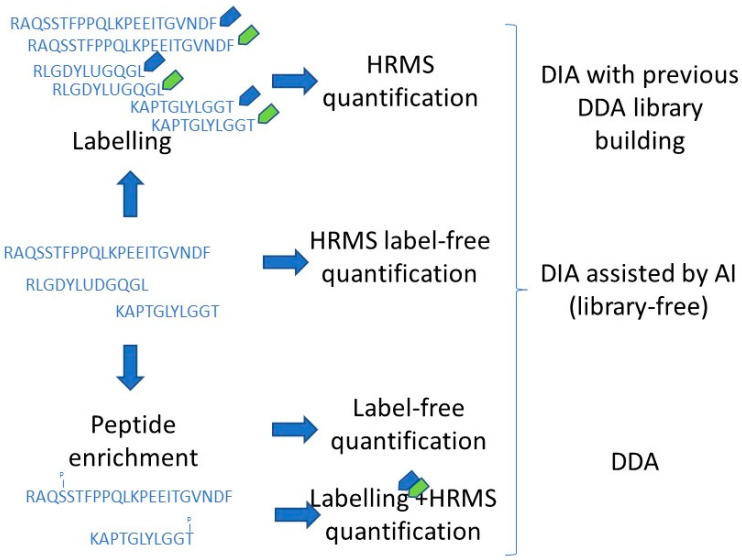
Workflow of labelling and label-free mass spectrometry-based proteomics, as well as peptide enrichment for detecting post-translational modifications. Blue and green tags mean isobaric labelling. HRMS: high resolution mass spectrometry; DIA: data-independent acquisition; DDA: data-dependent acquisition; AI: artificial intelligence.

**Table 1 ijms-24-04709-t001:** Some important foodborne mycotoxins, commodity of origin, main producing moulds, and toxic effects on human health [29,30,31,32,33,34,35].

Mycotoxins	Major Foods	Mould Sources	Toxic Effects
Aflatoxins: B_1_, B_2_, G_1_, G_2_,	Cereals, nuts, spices	*Aspergillus* section Flavi	Hepatotoxic, immunosuppressive, carcinogenic, teratogenic, genotoxic
Ochratoxin A	Cacao, dried fruits, wine, cereals, spices, dry-cured meats	*Aspergillus* section *Circumdati* *Aspergillus* section *Nigri* *Penicillium verrucosum*, *P. viridicatum*, *P. nordicum*	Carcinogenic, teratogenic, genotoxic, immunotoxic
Fumonisins: B_1_, B_2_	Maize	*Fusarium* section *Liseola*	Carcinogenic, pulmonary oedema, neurotoxic, cardiovascular toxicity
Patulin	Apple	*P. expansum*, *Aspergillus clavatus*, *Bysochlamis nivea*	Acute toxicity, neurotoxic, genotoxic, carcinogenic, teratogenic, immunotoxic
Trichothecenes: T-2, DON, DAS, HT-2, NIV, etc. ^a^	Cereals	*Fusarium acuminatum, F. poae, F. sporotrichioides, F. graminearum, F. colmorum, F. cerealis*	Vomiting, diarrhoea, leukopenia, necrotic lesions, haemorrhage, kidney problems, immunosuppressive
Zearalenone	Cereals	*F. graminearum, F. culmorum, F. equiseti, F. cerealis, F. verticillioides, F. incarnatum*	Oestrogenic effects, reproductive toxicity
*Alternaria* mycotoxins: AOH, AME, ALT, TeA, etc. ^b^	Tomato, sunflower seed, cereals	*Alternaria* sp.	Acute toxicity, cytotoxic, fetotoxic, teratogenic, haematological disorders, oesophageal cancer

^a^ T-2: toxin T2; DON: deoxynivalenol; DAS: diacetoxyscirpenol; HT2: toxin HT2; NIV: nivalenol. ^b^; AOH: alternariol; AME: alternariol monomethyl ether; ALT: altenuene; TeA: tenuazonic acid.

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
