# Peer review of "Proteomics as a New-Generation Tool for Studying Moulds Related to Food Safety and Quality"

_ijms, 2023, doi:10.3390/ijms24054709_

Round 1

Reviewer 1 Report

The manuscript entitled " Proteomincs as new-generation tool for studying moulds related to food safety and quality" is of huge interest because gives an overview of all the recent advances in the field.

General comments: The manuscript is very weel organized. The information is very complete. The presented tables make the manuscript easier to follow. The weakest part of the manuscript is the abstract because in my opinion it does not summarize the content. Besides, the introduction  section could also be improved. Moreover other ways of avoiding moulds growth in food could be given.

As far as I concern, the manuscript could be accepted after these minor improvements and the english carefully check.

Best regards!

Author Response

The authors thank the Reviewer for his/her comments. Most of suggestions have been followed, excepting that associated with strategies to avoid the growth of foodborne moulds since they are mentioned, and some of them developed, in section 2 (lines 155-185). As also stated by the Reviewer, the English grammar has been deeply revised.

Reviewer 2 Report

The manuscript entitled "Proteomics as new-generation tool for studying moulds related to food safety and quality" needs revision. Please refer to comments given in the text of reviewed attached file of the manuscript. 

Author Response

The authors thank the Reviewer for his/her comments. All the suggestions included in the attached file have been followed and are marked up using the “Track Changes”.

Reviewer 3 Report

I enjoyed reading the article. It is well-structured and has a lot of summarized information. 

Author Response

The authors thank the Reviewer for his/her comments.

Reviewer 4 Report

Minor comments

1. Development of resistant variety through genetic engineering is an effective approach to combat negative effects of moulds. Authors need to discuss about genetic modifications in food products with some suitable references.

2. Sequencing techniques should be discussed to better understand the genetic background of moulds.

3. The work flow of proteomics methodology should be included in the manuscript.

4. A diagrammatic representation for proteomic analysis of the food associated microorganisms should be included.

5. Line number 201-204 seems to be irrelevant better to remove it.

6. Line number 335-338 is confusing and requires editing.

Author Response

The authors thank the Reviewer for his/her comments.

  1. Development of resistant variety through genetic engineering is an effective approach to combat negative effects of moulds. Authors need to discuss about genetic modifications in food products with some suitable references.

Response: As far as we are concerned, genetic modifications must not be included since the review is based on Proteomics and not on Genomics or Transcriptomics.

  1. Sequencing techniques should be discussed to better understand the genetic background of moulds.

Response: As considered in the previous comment, it does not make sense to include genetic information.

  1. The work flow of proteomics methodology should be included in the manuscript.

Response: As the Reviewer suggests, the flow diagram has been added to the revised version of the manuscript. Please, see Figure 1.

  1. A diagrammatic representation for proteomic analysis of the food associated microorganisms should be included.

Response: As the Reviewer suggests, a representation for proteomic analyses has been added to the revised version of the manuscript. Please, see Figure 2.

  1. Line number 201-204 seems to be irrelevant better to remove it.

Response: As requested by the Reviewer, the sentence has been removed.

  1. Line number 335-338 is confusing and requires editing.

Response: The sentence has been rewritten to guarantee its understanding.

Reviewer 5 Report

-Table 1. Include the chemical structures of main mycotoxins

-Include an image as example of proteomic metodology to find mould growth and/or mycotoxin production

Author Response

The authors thank the Reviewer for his/her comments.

-Table 1. Include the chemical structures of main mycotoxins.

Response: Since 18 different mycotoxins are mentioned in Table 1, we considered that it is not feasible to include all the chemical structures. Besides, this information is not relevant for the aim of the review.

-Include an image as example of proteomic metodology to find mould growth and/or mycotoxin production.

Response: Considering this suggestion together with that of Reviewer 4, Figures 1 and 2 have been included. They are more complete than the image requested by Reviewer 5 due to the fact that they schematize the methodology to be followed for evaluating the mechanisms of action beyond the mould growth and mycotoxin production inhibition. Apart from that, it has to be taken into account that Proteomics can be used to identify the mould diversity in a microbial community and study the mechanisms of action of antifungal treatments, but not to detect mould growth or mycotoxin production.